# Inflammatory Disorders of the Central Nervous System Vessels: Narrative Review

**DOI:** 10.3390/medicina58101446

**Published:** 2022-10-13

**Authors:** Aleksandra Ekkert, Marta Šaulytė, Dalius Jatužis

**Affiliations:** 1Centre of Neurology, Vilnius University, Santariskiu 2, LT-08661 Vilnius, Lithuania; 2Faculty of Medicine, Vilnius University, LT-08661 Vilnius, Lithuania

**Keywords:** central nervous system vasculitis, PACNS, amyloid-related brain inflammation, systemic vasculitis neurologic manifestations

## Abstract

Inflammatory disorders of the central nervous system (CNS) vessels, also called CNS vasculitides, can cause substantial disability or even be fatal. Inflammation of the CNS vessels can be caused by primary angiitis of the CNS (PACNS), inflammatory cerebral amyloid angiopathy, or systemic inflammatory disorders. Clinical symptoms of these disorders are often non-specific, such as encephalopathy, cognitive and affective abnormalities, headache and focal neurological symptoms. Diagnostic workup includes a thorough neuropsychiatric examination, blood and cerebrospinal fluid analysis and magnetic resonance imaging (MRI) of the brain and its vessels. Biopsy of the brain remains the gold standard diagnostic test. Timely diagnosis and treatment initiation is of high importance, as it might prevent severe complications, such as ischemic and hemorrhagic stroke. In this review, we describe the specific characteristics of primary and secondary non-infectious CNS vasculitides which help to establish the diagnosis, discuss the peculiarities of the diagnostic workup and present current treatment recommendations.

## 1. Introduction

Central nervous system (CNS) vasculitides comprise a heterogenic group of disorders that can cause substantial disability or even be fatal if the treatment is not applied in a timely fashion [1]. Acquired lesions of the CNS vessels can be caused by primary angiitis of the CNS (PACNS), inflammatory cerebral amyloid angiopathy (ICAA), systemic inflammatory disorders, infectious agents, drugs, radiation and possibly chronic traumatic encephalopathy [2,3,4]. This review article is focused on inflammatory non-infectious CNS vasculopathies or CNS vasculitides.

Inflammatory CNS vasculopathies have been described in the literature relatively recently. Clinical cases showing clear evidence of cerebral involvement in systemic vasculitis were brought to light only approximately a hundred years ago [3]. Due to the lack of knowledge of the pathogenetic mechanisms of the disorder, treatment options were also scarce, resulting in poor prognosis of some subtypes until the end of the 20th century [5]. As a consequence of non-specific typical manifestations, such as headache and encephalopathy [6], timely diagnosis and treatment become delayed, resulting in further complications and a worse prognosis [7].

Inflammatory disorders of brain vessels can be primary (PACNS, ICAA) when the brain is the only or the most affected site, or secondary caused by systemic vasculitides and connective tissue diseases. Secondary vasculitides are classified according to the 2012 revised International Chapel Hill Consensus Conference Nomenclature of Vasculitides [8]. Table 1 shows the classification of the vasculitides affecting the nervous system.

CNS vasculitis should be differentiated from mimicking disorders, such as infectious or immune-mediated encephalitis, neoplastic and demyelinating lesions and reversible cerebral vasoconstriction syndrome (RCVS) [9,10]. The workup process is frequently complicated by the lack of specific laboratory and radiological findings [11]. As a result, a broad diagnostic workup is recommended, which should include a thorough neuropsychiatric examination, blood and cerebrospinal fluid analysis, ultrasound examination of the head and neck vessels, magnetic resonance imaging (MRI) of the brain, magnetic resonance angiography (MRA), computed tomography angiography (CTA) and digital subtraction angiography (DSA) of the head and neck vessels [6,9,12,13,14]. Other sophisticated methods, such as fluorodeoxyglucose positron emission tomography (FDG-PET) and vessel wall imaging (VWI) are also recommended, although not applied widely so far [9]. Brain biopsy is also highly recommended, remaining the gold standard for the CNS vasculitis workup, as the results of the conventional non-invasive tests are frequently inconclusive.

This paper aims to highlight specific clinical, laboratory and imaging features of different types of inflammatory CNS vasculopathies and provide some insights on the differential diagnosis of non-infectious CNS vasculitides and treatment guidelines for the non-infectious CNS vasculitides.

## 2. Literature Search and Synthesis

A search of scientific articles was performed in PubMed, ClinicalKey and UpToDate databases, using the keywords “CNS vasculitis”, “primary CNS vasculitis”, “primary CNS angiitis”, “CNS vasculopathy”, “inflammatory cerebral amyloid angiopathy”, “neuropsychiatric lupus”, “lupus CNS vasculitis”. The authors used the filter to select papers published within the last 10 years. Nevertheless, older publications were also taken into the consideration, if fundamental and highly cited. This literature review includes 54 scientific publications describing primary central nervous system angiitis, inflammatory cerebral amyloid angiopathy, central nervous system vasculitides caused by systemic vasculitis and connective tissue diseases. Figure 1 illustrates the process of literature analysis.

## 3. Primary Vasculitides of the CNS

### 3.1. Primary Angiitis of the Central Nervous System

Primary angiitis of the central nervous system (PACNS) is a rare inflammatory vascular condition of unknown etiology that affects only the blood vessels in the central nervous system (CNS)—the brain, spinal cord and meninges [1,2,3,4,5,6,7,8,9]. People of all ages are affected, but the highest incidence is found in the sixth decade of life [1,2,4,5,6,7,8,9,10]. Some studies describe similar incidence between both sexes [1,6,10], while other authors highlight a slight male predominance [2,4,8,9]. It is problematic to estimate the true incidence of the disease due to its rarity and lack of epidemiological studies, but the available data suggests an annual incidence rate of 2.4 cases per 1 million person-years [1,5,7,11].

The exact pathogenesis of PACNS remains obscure [13]. It is thought that specific activation of the immune system, specifically T cells. due to various triggers causes the inflammation of blood vessels in the CNS leading to narrowing or blockage resulting in significant ischemia and necrosis [15,16]. Acute PACNS frequently presents as an ischemic or hemorrhagic stroke leading to new focal neurological symptoms, none of which occurs in more than 25–30% of patients [16,17]. There are three histological subtypes of PACNS: granulomatous (56–58%), lymphocytic (20–28%) and necrotizing (14–22%), the last subtype frequently presenting with intracerebral hemorrhage [7,18].

PACNS does not have a pathognomonic clinical picture and may present with a range of non-specific symptoms [5,13,14,19,20]. The most common initial symptoms are headache (50–60%), confusion, cognitive impairment and behavioral and personality changes (50–70%) [15,17,18]. Unfortunately, there are no unequivocal headache characteristics that would help suspect PACNS [6]. Based on the size of the affected blood vessels, two subtypes of PACNS are recognized: the small-vessel variant and the large/medium-sized vessel variant [15,18,21]. Patients affected by the small-vessel subtype encounter seizures, dyskinesias, consciousness and cognitive impairment more frequently, while patients with large and medium vessel involvement tend to present more with focal deficits [21]. Isolated stroke within a single blood vessel, in the absence of symptoms of diffuse brain damage (drowsiness, confusion) and unexplained headache, is extremely rare [5]. Systemic symptoms such as weight loss, night sweats, fever, or rashes are not associated with PACNS [5,13,14,17]. Most patients have a long prodromal period with headaches and mild cognitive changes, lasting from a few weeks to several months [14]. Approximately 40% of patients seek medical attention more than 3 months after the occurrence of the first symptoms and another 40% visit the doctor within 4 weeks [16,17,20].

In 1988, Calabrese and Mallek proposed that PACNS should always be suspected when all three diagnostic criteria are fulfilled [22]: (1) acquired neurological deficit whose cause remains unknown after a thorough investigation; (2) typical angiographic or histopathological findings of vasculitis in the CNS; (3) no evidence of other conditions that could cause the aforementioned findings.

Since there are no tests with sufficient sensitivity or specificity for PACNS, serologic tests are mainly useful to rule out alternative diagnoses, such as infection, other systemic autoimmune diseases, or malignancy [6,12,13,14]. Erythrocyte sedimentation rate (ESR) and C-reactive protein (CRP) are typically normal as the inflammatory process is restricted to the CNS [6,13,14,16]. Recommended laboratory and radiological workups necessary to differentiate vasculitis mimics are listed in Table 2.

Cerebrospinal fluid (CSF) is abnormal in about 90% of histologically proven vasculitis cases, so lumbar puncture should be performed in all patients with suspected PACNS [13,14,15,16]. The majority of patients (80–90%) show mild to moderate lymphocytic pleocytosis (about 10–20 cells/mL), elevated protein levels (about 120 mg/mL) and normal glucose levels [6,12,14,15,17].

Alterations on MRI are observed in more than 90% of cases of PACNS and, although its specificity is low, this test is highly sensitive for initial evaluation [1,5,12,14,15,16,17,19,20]. Ischemic changes are the most common, found in up to 53% of cases [1,15]. Multifocal cortical and subcortical infarcts are detected, often bilateral, with contrast-enhanced parenchymal and meningeal areas, intracranial hemorrhages and enhanced signal areas on FLAIR and T2-weighted sequences [1,14,19]. Other common findings include diffuse small-vessel changes and confluent white matter changes, which can often be confused with multiple sclerosis or cortical necrosis [19]. The incidence of parenchymal hemorrhages in PACNS varies from 8% to 55% [5,14]. Tumor-like lesions with mass effect are observed in a small proportion of patients (5–10%) [1,5,19]. Even generalized atrophy on MRI can be a sign of PACNS [6]. Very rarely, PACNS involves only the spinal cord and such a finding must be confirmed by biopsy [5]. Cerebrospinal fluid anomalies and gadolinium enhancement on MRI are more typical for the small-vessel subtype [21].

Direct or indirect cerebrovascular imaging using conventional angiography, computed tomography angiography (CTA) or magnetic resonance angiography (MRA) can detect areas of intermittent vasoconstriction and dilatation, resembling a beaded necklace [1,5,13,15,19]. However, specificity for vasculitis is only 30%, as such vascular abnormalities are common to many other pathologies [5,17,19,20]. Other angiographic findings include tapering of the vessel lumen of a single or many vessels and fusiform arterial dilatations, multifocal vascular occlusions, development of collateral circulation, or delayed contrast-medium enhancement and washout time [1,19]. Unfortunately, angiography has limited resolution for small vessels so the diagnosis of PACNS cannot be ruled out in the absence of lesions on angiography [12,14,17].

The gold standard for diagnosing PACNS is brain biopsy [5,13,14,16,18,19,24], showing a clear diagnosis of vasculitis or other specific pathology in approximately 75% of cases [17,20]. It has excellent specificity for PACNS. However, the sensitivity of biopsy varies from 50% to 83%, which is attributed to both the patchy nature of the disease and biopsy techniques [1,14,16,17,18,20]. Some clinicians are concerned about the risks of such an invasive procedure, especially when the biopsy must be taken from the dominant temporo-parietal region, brainstem, or spinal cord. However, the risk of complications from brain biopsy is less than 1% [17,19,20], while the risk of immunosuppressive treatment is even higher [13,20].

Due to the rarity of PACNS, current treatment recommendations are based on case reports, retrospective observational studies and expert opinion. Due to the inflammatory and autoimmune nature of PACNS, treatment strategies focus on immunosuppressive therapy [13], based on the relevant principles for the treatment of systemic vasculitis [5,14,18]. The most commonly used are: glucocorticoid (prednisone) monotherapy or combination therapy of glucocorticoids and cyclophosphamide [5,14,20], both showing similar response rates of approximately 80% [15].

Treatment with oral prednisolone should be started immediately at the time of diagnosis at an initial dose of 1mg/kg/day or 60mg/day (but not more than 80mg/day) until stabilization is achieved, and then tapered gradually over a period of months [5,13,15]. For most patients, 12–18 months is an adequate treatment duration [15]. In more severe cases, intravenous methylprednisolone pulse therapy might be used (1g intravenously daily for 3–5 days), followed by oral prednisolone (1mg/kg/day) for 1 month, then slowly tapered over 12 months [15,16,18].

The decision to use combination therapy with cyclophosphamide is based on the pace of the disease and the severity of neurological symptoms [5]. Cyclophosphamide should be added if no positive response is obtained after initial glucocorticoid monotherapy [13]. Cyclophosphamide can be used orally as a 2 mg/kg daily dose or monthly intravenous pulse therapy (starting at 750 mg/m^2^). For severe adverse effects, including infection, cystitis, bladder cancer and gonadal toxicity, cyclophosphamide should usually be administered in collaboration with a rheumatologist or other physician with experience in the use of this drug and in monitoring adverse effects [17]. To optimize safety, regular monitoring of kidney, liver and bone marrow functions is essential. The duration of cyclophosphamide treatment is usually 3 to 6 months, after which the patient may be considered to be in remission if the disease has not progressed clinically or radiologically. Given the high rate of improvement with glucocorticoid monotherapy or combination therapy with cyclophosphamide in long-term follow-up cohorts (~80%), the validity of the diagnosis should be reassessed if patients are considered to have PACNS failed induction therapy. In patients with credible diagnoses, the options are limited [5].

Once remission is achieved, after about 6 months, patients might switch to the maintenance regimen using long-term immunomodulating treatment, such as azathioprine, mycophenolate mofetil, or methotrexate [5,16,17,18]. In patients receiving maintenance treatment, the relapse rate was found to be lower (21%) compared with patients who did not receive it (41%) [14]. Clinical, CSF and neuroradiological monitoring are necessary to determine the individual duration of maintenance treatment [18]. Periodic MRI and MRA examinations are recommended every 4–6 weeks after the start of treatment and then every 3–4 months during the first year to monitor disease progression [5,15]. The relapses are treated with intravenous and oral glucocorticoids, e.g., methylprednisolone 500–1000mg pulse therapy for 3–5 days [18]. If PACNS is resistant to glucocorticoids and immunosuppressants, tumor necrosis factor alpha blockers (infliximab, etanercept) and mycophenolate mofetil can be used as adjunctive therapy to standard treatment [15].

### 3.2. Inflammatory Cerebral Amyloid Angiopathy (ICAA)

ICAA is an increasingly recognized condition characterized by an inflammatory response to the amyloid β deposits in small and medium-sized arteries and choroidal vessels of the brain [16,25]. This condition is divided into two types based on pathological features: cerebral amyloid angiopathy-related inflammation (CAARI), characterized by perivascular inflammatory infiltration and amyloid β-related angiitis (ABRA), causing transmural inflammatory process in the vessel, with or without formation of granuloma [26]. Both variants can be clinically quite similar to PACNS but differ in characteristic radiological features [27]. Both CAARI and ABRA are mostly detected in patients aged 60 years and older, which is typically younger compared to non-inflammatory cerebral amyloid angiopathy patients (mean age 76 years), but older than PACNS patients (mean age 45 years) [16,28,29]. ICAA incidence does not differ in men and women; the exact prevalence is unknown [25,26,27,30].

ICAA mechanism is still unrecognized. Why amyloid deposition is accompanied by the inflammatory response only in certain subjects is a subject of particular interest. Some authors state that APOE ε4/ε4 genotype could be linked to the increased risk of the inflammatory response [7]. ICAA has been linked to other autoimmune disorders, such as hypothyroidism, Graves’ disease, rheumatoid arthritis and autoimmune hepatitis [29].

The clinical presentation consists of acute or subacute cognitive decline, focal or multifocal neurological symptoms, often mimicking stroke, as well as new new-onset seizures or headaches [27,28]. The clinical course can vary widely, from acute to chronic, in some cases lasting many years [27]. Contrarily to PACNS, cognitive decline is the most common symptom, in both CAARI (presenting in 48–76% of patients) and ABRA (presenting in about 71% of patients) and may present with confusion, memory loss, apraxia, impairment of executive functions or decision-making, eventually progressing to severe dementia and behavioral disorders [16,25,26,28,29]. The second most common symptom (CAARI—46%, ABRA—51%), is a focal neurological deficit, such as hemiparesis, sensory loss, aphasia, hemianopsia, or ataxia [16,27,28,29]. Another common symptom (32–41%) is severe headache [25,28,29]. Dizziness, nausea, or vomiting may also be present, although general constitutional signs are usually not significant and occur rarely [27]. A small number of patients may experience hallucinations [30]. Seizures occur in about one-third of patients (31%) [16,28,29]. In severe cases, when there is a disturbance of consciousness, status epilepticus should be considered [27]. Clinical characteristics of primary CNS vasculitides are compared in Table 3.

As in PACNS, no specific laboratory or serological tests exist [27,29]. Most patients have elevated inflammatory markers, but their diagnostic value is insignificant [16,26,27,29]. Many patients have CSF abnormalities, such as elevated protein levels (>45 mg/dl) and mild lymphocytic pleocytosis (>5 cells/mm3) [25,26,27,28,29]. Up to 70% of ABRA patients have the ApoE e4/e4 genotype [29]. Another important finding is amyloid β antibodies in the CSF [25,26]. Some authors imply that, in the future, the detection of amyloid β antibody levels might be useful for monitoring the response to treatment [25,32].

CAARI-specific MRI changes include symmetric or asymmetric, punctate or confluent T2/FLAIR hyperintense lesions in the cortex or subcortical white matter, as well as microhemorrhages, a specific characteristic of non-inflammatory cerebral amyloid angiopathy [27,28,29]. More than half (53.5%) of CAARI patients have cortical or subcortical edema, which is detected by both CT and MRI [26,30]. The edema is usually confluent, which allows differentiation from PACNS. Other non-specific white matter changes, such as infarcts or atrophy, may also occur [30]. ABRA-specific MRI findings are asymmetric T2/FLAIR hyperintense lesions with minimal contrast enhancement, extending into the juxtacortical regions [16,29]. Gradient echo (GRE) and susceptibility-weighted imaging (SWI) sequences detect microhemorrhages, which are usually located in the corticospinal junction and temporo-occipital region [29]. Conventional angiography usually does not show abnormalities, as these disorders involve lesions of small vessels that are not visible [16]. As both CAARI and ABRA imaging abnormalities are quite similar, definitive diagnosis is only confirmed by biopsy [30]. As the treatment of both disorders is similar, differentiation between the two has more of a theoretical significance.

Periodic brain MRI scans are appropriate to monitor the response to treatment, as ICAA lesions tend to improve with treatment, although clinical amelioration may precede radiological improvement [28,30]. Relapses are characterized by the recurrence of T2/FLAIR hyperintensity, often in the same area that was previously affected. It is worth noting that the volume of lobular T2/FLAIR lesions may decrease with immunosuppressive therapy or even spontaneously, but the periventricular hyperintensity reflecting the burden of non-inflammatory lesions remains unchanged [27].

Recommendations for the treatment of inflammatory cerebral amyloid angiopathy are based on a case-by-case basis, small series reports and much of the experience with PACNS [16,27]. Treatment usually starts with an intravenous methylprednisolone pulse therapy of 1g for 5 days, followed by oral prednisolone 1mg/kg/d, which is gradually reduced over 4–8 weeks. In ABRA patients and severe CAARI cases, cyclophosphamide 750 mg/m^2^ intravenous infusion every month for 3–6 months or 15 mg/kg (maximum 1.2 g) every 2 weeks for the first three pulses, followed by infusions every 3 weeks for the remaining 3–6 pulses can be added [16,27]. Cyclophosphamide is associated with better and faster control of ABRA [27]. In the absence of response to treatment or in case of early relapse an oral mycophenolate mofetil, methotrexate, azathioprine, rituximab, or intravenous immunoglobulin should be considered [16,27,28,29]. Treatment is effective in approximately 72% of patients. The clinical course is variable, but most patients recover within a few weeks [28]. The exact duration of treatment has not yet been established and, in most cases, long-term treatment is necessary as the disease may relapse. After clinical stabilization, it is suggested that the treatment should be continued for at least one year [27,28]. In case of relapse, patients usually respond well to immunosuppressive therapy [27].

## 4. Secondary Vasculitides of the Central Nervous System

### 4.1. Central Nervous System Involvement in Systemic Vasculitis

#### 4.1.1. Giant Cell Arteritis

Giant cell arteritis (GCA) is a chronic form of vasculitis of large and medium-sized blood vessels, characterized by a granulomatous inflammatory process affecting the aorta and its main branches, mainly the extracranial branches of the carotid artery, especially the temporal artery [24,33]. It is the most common primary systemic vasculitis, with an overall incidence of 15–25 cases per 100,000 person-years [33]. The disease is most common in people over 50 years of age and twice as commonly affects women as men [9,33,34,35]. The frequency of CNS involvement in GCA and other systemic vasculitides is shown in Table 4.

The disease usually starts gradually over a few weeks or months, but in up to 20% of patients, it may start suddenly [34]. The vast majority of patients with GCA complain of sudden-onset, persistent and very intense headaches, usually in the temporomandibular or occipital regions, but it can be generalized to the face, jaw and neck [33,34]. On physical examination, the superficial temporal artery feels thickened and knotted, characterized by tenderness and reduced pulsation [33,34]. Approximately half of the patients have systemic symptoms such as febrile fever, night sweats, anorexia and weight loss [33]. A typical feature of GCA is an aortic arch syndrome, characterized by hand claudication, pulse asymmetry and paresthesias. Jaw claudication might be present, which is aggravated by speaking or chewing [33]. About 15% of patients develop ophthalmic complications, including ischemic optic nerve neuropathy and blindness. The loss of vision is painless and usually irreversible [34].

Stroke has been reported to occur in 1–3% of GCA patients, including cerebral infarction in 58%, subarachnoid hemorrhage in 24% and intracerebral hemorrhage in 18% [35]. Ischemic strokes occur due to acute thrombosis, micro-embolism, or atherosclerotic vascular lesions caused by chronic inflammation [34,35].

The gold standard for diagnosing GCA is a temporal artery biopsy, which detects mononuclear cell infiltration or granulomatous inflammation with giant cells [33,34]. Even with a negative biopsy result, GCA cannot be ruled out because of the patchy distribution of lesions. Laboratory results usually show an increase in ENG and CRB, anemia and thrombocytosis [33]. Ultrasonography shows hypoechoic, thickened superficial temporal artery wall and halo sign (Figure 2). CTA and MRA show thickening of the vessel wall, stenosis and occlusions [9]. Glucocorticoids are the main treatment of choice [33].

#### 4.1.2. Takayasu Arteritis

Takayasu arteritis (TA) is a rare chronic inflammation of large blood vessels involving the aorta and its main branches and pulmonary arteries [9]. The prevalence of TA is much lower than that of GCA [24]. The disease occurs in people younger than 40 years of age, predominantly in Asian women (F:M = 10:1). Granulomatous inflammation of the vessel wall with intimal fibrosis and proliferation causes vascular narrowing, occlusion and aneurysm formation [9,24,35].

Clinical signs at the onset of the disease may be non-specific and include fever, fatigue and joint pain. Eventually, intermittent claudication of the arms or legs due to circulatory disturbances, Raynaud’s syndrome, angina pectoris and differences in pulse or blood pressure between limbs appear [24]. More than half of patients develop neurological symptoms ranging from headache/dizziness to ischemic or hemorrhagic stroke. Headache or dizziness are the most common neurological complaints, occurring in 25.5–78.1% of patients, with visual disturbances in up to 60% of patients and syncope and TIA in about one-fifth. Stroke, the most serious complication of TA, occurs in 10–20% of cases, the majority of which are ischemic, usually due to multiple and severe stenotic or occlusive lesions in the aortic arch and its main branches. Other rare neurological complications may include posterior reversible encephalopathy syndrome (PRES), Horner’s syndrome, unilateral sensorineural hearing loss, multiple cranial nerve palsies and brachial plexus injury due to brachial artery aneurysm [35].

TA patients usually have elevated ESR and CRP [24]. CTA and MRA have high sensitivity and specificity (>90%). Vascular wall thickening and contrast accumulation in the early phase of MRA correlates with an increase in markers of inflammation. In addition, vascular wall edema on T2-weighted MRI sequences and increased carotid artery intima-media thickness on ultrasound may also be markers of active inflammation [9]. Glucocorticoids are routinely prescribed for the treatment of TA, but other immunosuppressive agents may be considered [40].

#### 4.1.3. Nodular Polyarteritis (Polyarteritis Nodosa)

Nodular polyarteritis (NPA) is a rare systemic necrotizing vasculitis, mainly affecting medium and small arteries that can affect almost all organ systems [35,36]. The disease is more common in older men [9].

NPA causes micro-aneurysms, stenoses and thromboses, leading to ischemia or hemorrhage of the tissues supplied by the affected vessels [35]. Common clinical symptoms include fever, muscle and joint pain, renal failure, skin involvement such as purpura, livedo reticularis and skin nodules, heart failure and myocardial infarction and gastrointestinal complaints with characteristic abdominal pain due to visceral infarcts [24,36].

PNS damage occurs in 60–70% of cases and usually becomes clinically evident early in the course of the disease, typically within a few months. PNS lesions can present as mononeuritis, polyneuritis, multiple mononeuritis and symmetrical peripheral neuropathy [35]. The combination of polyneuropathy and livedo reticularis should raise a strong suspicion of NPA [24]. CNS involvement is present in 5–25% of NPA cases and is more common in late-stage disease [35,36]. CNS involvement is associated with a poor prognosis, with an expected 5-year mortality of 26% [35]. The main focal CNS disorders are ischemic stroke (13–17%) and intracerebral and subarachnoid hemorrhage [35,36]. The most common manifestations are encephalopathy, seizures and focal neurological symptoms, sometimes mimicking multiple sclerosis [24,35]. Patients with nodular polyarteritis may also develop pachymeningitis, characterized by headache and cranial nerve palsy [35].

There are no specific serological tests to diagnose the disease. In the case of CNS involvement, small cortical and subcortical infarcts are observed. Skin biopsies reveal polynuclear deposits in the walls of small and medium-sized blood vessels. Angiographic studies are also useful, as they show segmental dilatations and constrictions of blood vessels [36] The disease is usually treated with glucocorticoids, but a more aggressive course of treatment with glucocorticoids and cyclophosphamide is required in the presence of CNS or other vital organ involvement [41].

#### 4.1.4. Kawasaki Disease

Kawasaki disease is a systemic, predominantly medium-sized vasculitis affecting children up to 5 years of age, more often boys [35]. The highest incidence is in Japan [36]. Clinical symptoms include fever of at least 5 days duration, erythema or edema of the extremities, polymorphic exanthema, bilateral conjunctivitis, erythema of the lips and oral cavity, ‘raspberry’ tongue, neck lymph node enlargement and coronary artery aneurysms, which can lead to myocardial infarction [9,36].

Potential neurological complications of Kawasaki disease include encephalopathy, seizures, aseptic meningitis, ischemic stroke, intracranial hemorrhage, ataxia and cranial nerve palsies [35,36]. Rare CNS imaging findings include transient subcortical FLAIR hyperintense lesions, subdural hematomas, cerebral infarcts and parenchymal atrophy [9]. The disease is treated with aspirin, intravenous immunoglobulin and glucocorticoids [42].

#### 4.1.5. Granulomatosis with Polyangiitis

Granulomatosis with polyangiitis (GPA, Wegener’s granulomatosis) is a rare form of small-vessel vasculitis characterized by necrotizing granulomatous ANCA-associated vascular inflammation [35]. The disease affects both sexes equally, typically in patients aged 40–60 years, and has an average incidence of 0.9/100,000 person-years [24].

Neurological complications occur in 29–50% of GPA patients [35]. The majority of these are PNS involvement, with peripheral neuropathy being the most common (60%) [19,35]. The occurrence of peripheral neuropathies is associated with a higher number of affected organs, increased ANCA titers and more severe disease [13]. CNS abnormalities occur in 7–11% of GPA patients [19,35,41]. The most common symptom is headache, followed by pachymeningitis and cranial nerve damage, with nerves II, VI and VII being the most commonly affected [35]. Inflammation of small blood vessels in the brain can cause encephalopathy or seizures [36]. CNS lesions are usually ischemic, but may also be granulomatous or hemorrhagic [41]. Hypo-physitis is another complication whose clinical expression varies depending on the site of the lesion [35].

MRI shows infarcts, hemorrhages and T2-hyperintense lesions, and contrast-enhanced scans can show brain thickening and infiltration of the meninges [9,36]. Brain biopsy shows necrotizing granulomatous vasculitis [36]. Secondary CNS involvement due to vasculitis is usually considered a serious condition with high mortality complications and requires aggressive treatment with high doses of glucocorticoids in combination with cyclophosphamide or rituximab [19,41].

#### 4.1.6. Granulomatosis with Polyangiitis

Eosinophilic granulomatosis with polyangiitis (EGPA, Churg-Strauss syndrome) is an ANCA-associated necrotizing, granulomatous small-vessel vasculitis with eosinophilia accompanied by pulmonary impairment with severe asthma attacks as the main symptom [24,35,43]. EGPA is traditionally described in three clinical phases: the prodromal phase is characterized by asthma, nasal polyps and rhinosinusitis, the eosinophilic phase is characterized by peripheral eosinophilic tissue infiltration and the vasculitic phase includes the clinical symptoms of systemic vasculitides, such as cardiac, cutaneous, pulmonary and neurological damage [19].

Neurological involvement is found in up to 60% of patients. It is mostly manifested by peripheral neuropathy, which usually precedes the development of visceral damage [35]. Approximately half of EGPA patients present with multiple mononeuritis, characterized by dysesthesia, paresthesia and edema of the distal limbs, especially the lower limbs [35,36].

CNS involvement is rare in EGPA, accounting for only 6–10% of all cases [35,36,43]. Cerebral infarcts and intracerebral hemorrhages due to intracranial vasculitis are the most frequent CNS complications [35]. However, other complications such as subarachnoid hemorrhage, cranial nerve palsy, encephalopathy, seizures, headache, venous sinus thrombosis, spinal hemorrhage, damage to the meninges, or optic neuropathy may also appear [19,35].

Laboratory tests for EGPA show eosinophilia and c-ANCA antibodies in the blood and biopsies show eosinophilic and granulomatous infiltration of the vascular wall [24,35]. EGPA is treated with glucocorticoids, cyclophosphamide and azathioprine [43].

#### 4.1.7. Microscopic Polyangiitis

Microscopic polyangiitis (MP) is a rare multisystem small-vessel vasculitis characterized by non-granulomatous inflammation with little or no immune deposits in the vessel wall [35,36]. PNS is more frequently affected than CNS and accounts for 55–79% of all MP complications [35]. PNS injuries mainly manifest as multiple mononeuritis [36]. CNS involvement is found in up to 11% of patients and may include headache, seizures, intracerebral infarction or hemorrhages, subarachnoid hemorrhage, hypertrophic pachymeningitis, PRES and spinal cord involvement [35,36].

MP is strongly associated with ANCA, with up to 75% of cases found in MPO-ANCA [24,35]. Treatment includes glucocorticoids combined with another immunosuppressant such as cyclophosphamide, azathioprine, or rituximab [44].

#### 4.1.8. Henoch-Schoenlein Purpura

Henoch-Schoenlein purpura (HSP), also known as IgA vasculitis, is an immunologically determined systemic vasculitis of small blood vessels, usually affecting the skin, gastrointestinal tract, kidneys and joints [35,36]. The disease is most common in children and people below the age of 20 years. Usually, clinical diagnosis is based on a characteristic rash confirmed by biopsy [36].

PNS injuries include brachial plexus, facial, femoral, ulnar and fibular neuropathies, Guillain-Barre syndrome and multiple mononeuritis. CNS involvement manifests as headache, impaired consciousness, seizures, focal neurological symptoms and speech or visual disturbances [35].

#### 4.1.9. Behçet’s Disease

Behçet’s disease is a systemic vasculitis affecting a wide range of blood vessels, both arteries and veins, and of different diameters [24,39]. The disorder is more prevalent in people under 40 years of age and is equally common in both sexes, but is associated with a worse prognosis in men [45]. The disease can cause recurrent inflammatory disorders in almost any tissue, with oral and genital ulcers and uveitis being the most characteristic symptoms [35]. Other manifestations include arthritis, gastrointestinal lesions (chronic diarrhea, bleeding), urinary tract (glomerulonephritis, urethritis, orchitis, epididymitis) and pulmonary lesions (parenchymal infiltration, chronic obstructive pulmonary disease, pulmonary hypertension) [45].

Neurological damage is found in 3–30% of patients. PNS lesions are very rare and include sensorimotor neuropathies, multiple mononeuritis, autonomic neuropathy and Guillain-Barre syndrome [39]. The manifestations of the disease in the CNS are divided into parenchymal and non-parenchymal neurovascular types [24,39].

The parenchymal type is more common and severe [19]. It is characterized by lesions in both hemispheres causing various focal neurologic symptoms, including motor, sensory, visual and speech disturbances, seizures, parkinsonism and other movement disorders. Brainstem lesions, cranial nerve palsy, cerebellar disturbances and aseptic meningitis are also possible [24,39,45]. Less common disorders include subcortical dementia, stroke and transverse myelitis [35].

The neurovascular type usually manifests as cerebral venous thrombosis, mainly due to large-vessel endothelial cell activation and aneurysms and increased intracerebral pressure resulting in headache, focal neurological symptoms, seizures, impaired consciousness and sixth cranial nerve palsy [24,39]. Arterial damage can lead to aneurysms, dissections, intracranial hemorrhages and infarcts [24].

Approximately 30–50% of patients with neurological disorders have a relapsing course of disease and factors contributing to an unfavorable prognosis include brainstem or spinal cord damage, frequent relapses, early disease progression, residual neurological effects in remission and significant cerebrospinal fluid pleocytosis [19].

The parenchymal type is characterized by significant T2 hyperintense lesions in the brainstem and less pronounced lesions in the basal nuclei, corpus callosum, cerebral hemispheres and cranial nerves. Acute lesions may be characterized by restricted diffusion, while atrophy is common in the chronic phase. Non-parenchymal lesions include aseptic meningitis, venous thrombosis, dissection, occlusion, infarction, or aneurysm. Occasionally, the lesions are massive, mimicking tumors or abscesses [9]. In patients presenting with the severe disease with nervous system involvement, steroids with an immunosuppressive agent such as azathioprine, cyclophosphamide, methotrexate, or mycophenolate mofetil are used [19,45].

### 4.2. Central Nervous System Vasculitis Due to Systemic Connective Tissue Diseases

#### 4.2.1. Systemic Lupus Erythematosus

Systemic lupus erythematosus (SLE) is an autoimmune disorder characterized by a wide range of dysregulation of the immune system, leading to systemic inflammation and multiple organ damage [46]. The disease has a wide spectrum of clinical features and affects women more often [9,37]. In 90% of cases, the disease manifests as skin vasculitis, but it can also cause visceral involvement. Lupus vasculitis is more likely to occur during active disease and is accompanied by general inflammatory symptoms and laboratory parameters such as increased inflammatory markers, fever, fatigue and weight loss [37].

SLE often affects both central and peripheral nervous systems, as a syndrome collectively known as neuropsychiatric lupus erythematosus (NPLE) [38,47]. NPLE is a severe complication of SLE, associated with high mortality, and can occur in approximately 30–50% of patients [19,36,37,48,49,50]. The mechanism of the NPLE is still unclear. It is thought that a disruption of the blood-brain barrier allows peripheral blood antibodies and immune components to penetrate the CNS, leading to brain inflammation and damage [47]. The main histopathological finding of NPLE is non-inflammatory microangiopathy associated with cerebral microinfarcts and thrombosis [37]. Antiphospholipid antibody-induced thrombosis is the second important mechanism of ischemic brain injury. Histopathological studies show that cerebral vasculitis is rare, with an incidence of 7–13% [19,36]. Overall, the pathogenesis of the disease is multifactorial, involving various inflammatory cytokines, autoantibodies and immune complexes directed against specific brain antigens [19,48].

According to the nomenclature published by the American College of Rheumatology, NPLE can be divided into two types according to the presentation: (1) neurological syndrome, resulting from focal parenchymal damage to the brain, spinal cord and peripheral nerves and (2) diffuse neuronal damage, characterized by a neuropsychiatric syndrome [49,50]. The focal neurological syndrome presents with focal seizures, stroke (both ischemic and hemorrhagic), vasculopathy and PRES, whereas diffuse syndrome has a much more diverse clinical presentation, including memory impairment, anxiety, cognitive impairment and depression [38].

CSF findings include pleocytosis, elevated protein and IgG index and decreased glucose levels [37,48]. MRI might reveal multifocal infarcts, basal ganglia lesions, white matter T2/FLAIR hyperintensities, cerebral atrophy, edema, hemorrhages and intracranial calcifications [48,51]. Angiography detects stenoses and aneurysms, while on MRA images vascular wall thickening and intramural accumulation of contrast could also be observed [37]. However, diffuse neuronal damage is not always evident on conventional MRI. More sophisticated modalities include MRI diffusion tensor imaging (DTI) [52].

Immunosuppressive drugs, including glucocorticoids and disease-modifying antirheumatic drugs, are the mainstay of treatment [46].

#### 4.2.2. Rheumatoid Arthritis

Rheumatoid arthritis (RA) is chronic autoimmune arthritis characterized by peripheral joint and systemic inflammation, mainly affecting middle-aged women [53]. The most common neurological manifestation is multiple mononeuritis [36,53]. CNS involvement is rare and includes cerebral vasculitis, aseptic meningitis and CNS rheumatoid nodules. Subacute onset of neurological symptoms (headache, seizures and focal neurological symptoms) with episodic exacerbations is typical [53].

Low C3 blood levels are common and brain MRI usually reveals hyperintense white matter lesions on T2-weighted sequences [53]. Figure 3 shows changes in the right frontal lobe in a subject who presented with encephalopathy, tetra-paresis, elevated rheumatoid factor and poly-arthropathy. Brain biopsy revealed necrotizing granuloma consistent with rheumatoid arthritis. Other infectious and autoimmune causes were excluded.

Rheumatoid CNS vasculitis should be distinguished from infectious or atherosclerotic forms, as it responds very well to immunosuppressants. Treatment includes glucocorticoids and disease-modifying traditional synthetic and biological drugs [53].

## 5. Future Directions in the Workup of Inflammatory Disorders of the Central Nervous System Vessels

Establishing the diagnosis of inflammatory CNS vasculopathies can be challenging, especially when only conventional biomarkers and imaging techniques are used. Thus, it is hard to overestimate the current need for more specific biomarkers that would be useful not only for the establishment of initial diagnosis, but also for monitoring the activity of the disease, predicting its clinical course, outcomes and response to treatment [24]. Furthermore, the availability of non-invasive biomarkers could decrease the need for brain biopsy. Novel biomarkers are constantly investigated. It is already well-known that IL-17 levels tend to be higher in subjects with PACNS-associated stroke compared to non-inflammatory stroke, whereas an increased amount of intrathecal CD4+ lymphocytes is associated with ABRA [24]. More evidence has been gained that sophisticated imaging modalities that are relatively rarely used in clinical practice so far, such as high-resolution vessel wall MRI [54], MRI DTI, [52] and FDG-PET [12], could improve diagnostic accuracy contributing to earlier diagnosis and timely treatment initiation. FDG-PET is useful for distinguishing large- and medium-sized vessel inflammation, such as GCA and medium-vessel PACNS, although not precise enough in small vessel vasculitides [12]. DTI could improve the diagnostic accuracy in NPSLE with diffuse neuronal damage by revealing decreased fractional anisotropy of various brain regions on DTI. Moreover, subtle microstructural changes can be observed even in non-NPSLE without clinical signs of cognitive impairment [52]. Undoubtedly, there is still a long way to go before clinicians would be willing and able to use the biomarkers routinely. Nevertheless, constant education and sharing of the experience would lead to better recognition of the clinical patterns of vasculitides and could bring to light mild and underdiagnosed forms of the disorder, thereby increasing currently claimed rare prevalence.

## 6. Conclusions

CNS vasculitides are rare disorders that are quite complex to diagnose due to a lack of specific clinical symptoms and diagnostic features. If the diagnosis is not established early, disease progression can result in severe complications or even death. Contrarily, timely diagnosis and treatment initiation frequently leads to remission. Thus, it is important to share knowledge about the clinical features and workup recommendations. We would like to sum up this review by emphasizing several key points. First of all, if no conclusive diagnosis is obtained for a substantial time, it is always useful to consider CNS vasculitis as an option. Second, it is crucial to differentiate between inflammatory and infectious causes, as the treatment should be initiated as early as possible. Finally, brain biopsy remains the most specific and rather safe method for differential diagnosis that should be considered early, because early treatment initiation results in a better prognosis.

## Figures and Tables

**Figure 1 medicina-58-01446-f001:**
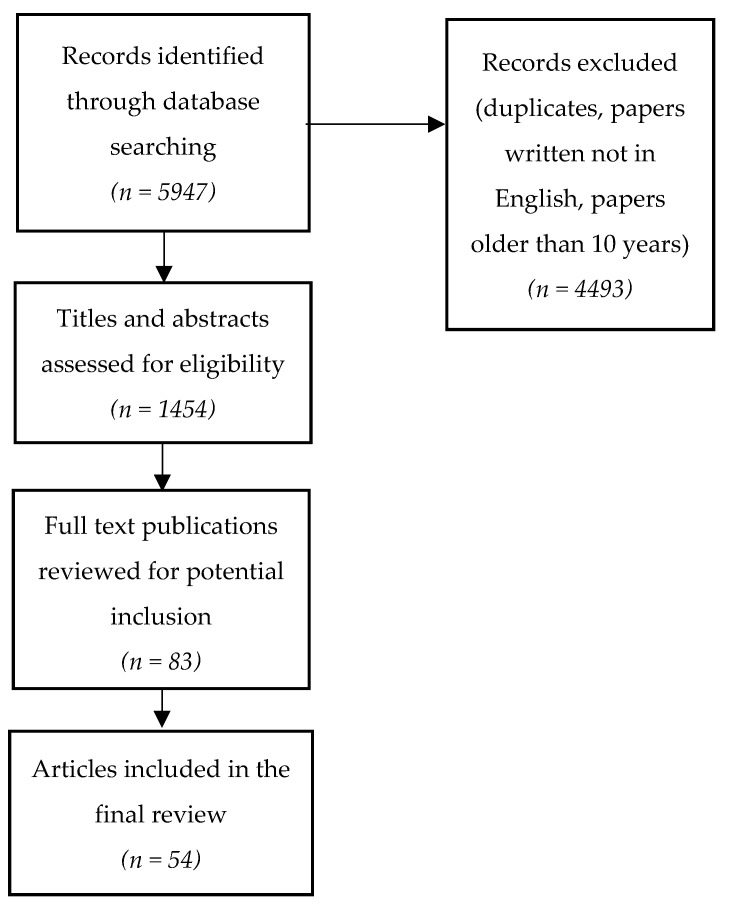
Literature search and analysis.

**Figure 2 medicina-58-01446-f002:**
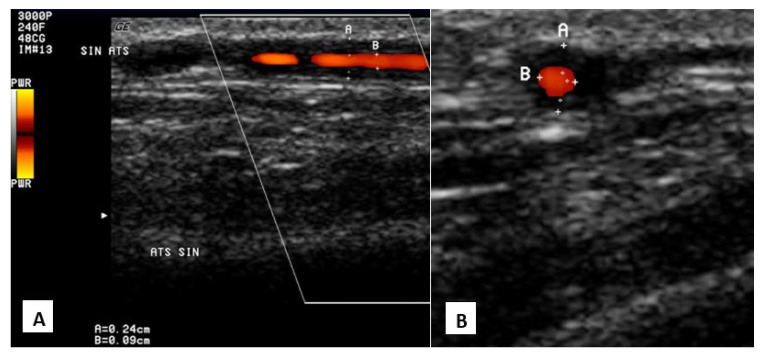
Giant cell arteritis in color-coded duplex ultrasound. “Halo” sign—circumferential hypoechoic vessel wall thickening around the lumen, attributable to vessel wall edema and intimal hyperplasia (superficial temporal artery, parietal ramus). A—longitudinal section, B—transverse section.

**Figure 3 medicina-58-01446-f003:**
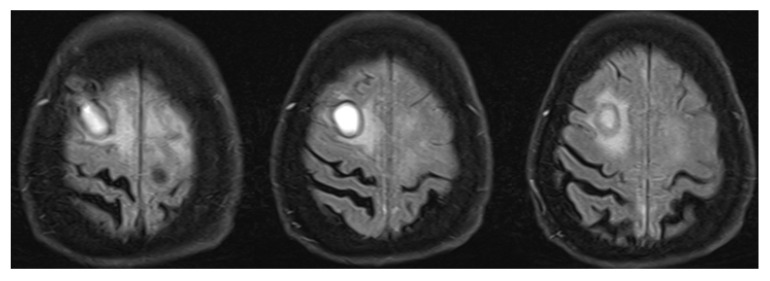
Right frontal lesion associated with rheumatoid arthritis. Right frontal lesion on T2 FLAIR MRI sequence that was differentiated between non-typical abscess and neoplastic changes (lymphoma, melanoma metastasis). Brain biopsy revealed necrotizing granulomatous inflammation with pachymeningitis, findings compatible with CNS lesion due to rheumatoid arthritis. T2 FLAIR—T2-weighted-fluid-attenuated inversion recovery; MRI—magnetic resonance imaging; CNS—central nervous system.

**Table 1 medicina-58-01446-t001:** Classification of the CNS vasculitides according to the affected vessel size and neurological manifestations.

Large vessel 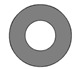	Takayasu arteritis Giant cell arteritis
Medium vessel 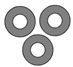	Polyarteritis nodosa Kawasaki disease
Small vessel 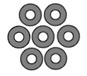	ANCA-associated: Microscopic polyangiitisGranulomatosis with polyangiitis (Wegener‘s granulomatosis)Eosinophilic granulomatosis with polyangiitis (Churg-Strauss syndrome) Immune complex vasculitis:Henoch-Schoenlein purpura
Variable vessel 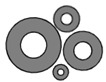	Behçet’s disease
Vasculitis associated with systemic disease	Lupus vasculitis Rheumatoid vasculitis
Single organ vasculitis	Primary angiitis of the CNS Inflammatory cerebral amyloid angiopathy

Modified from the 2012 revised International Chapel Hill Consensus Conference Nomenclature of Vasculitides [8]. ANCA—antineutrophil cytoplasmic antibodies; CNS—central nervous system.

**Table 2 medicina-58-01446-t002:** Clinical symptoms, laboratory and radiological features specific for the conditions mimicking PACNS.

Rheumatological causes [5,6,15,16,18].	Arthritis, skin changes, Raynaud phenomenon, ocular, lung, or kidney involvement Systemic symptoms Peripheral nervous system (PNS) involvement Elevated ESR and CRP Hypochromic anemia Low complement levels Elevated antibody titers: antinuclear antibodies extractable nuclear antigen antibodies antineutrophil cytoplasmic antibodies antiphospholipid antibodies rheumatoid factor
Infectious causes [5,6,10,15,16,18,23].	Positive serology for: Viral infections: varicella-zoster virus (VZV), human immunodeficiency virus (HIV), hepatitis C virus, cytomegalovirus, parvovirus B19 Bacterial infections: Treponema pallidum, Borrelia burgdorferi, Mycoplasma pneumoniae, Bartonella henselae, Rickettsia, Mycobacteria Fungal infections: aspergillosis, mucor-mycosis, coccidioidomycosis, candidosis
Malignancies (intravascular lymphoma, gliomatosis) [5,6,15,16,18].	History of malignancy Specific MRI spectroscopy findings Atypical cells on the brain biopsy
Reversible cerebral vasoconstriction syndrome [5,6,15,16,18].	Acute onset, thunderclap headache Normal cerebrospinal fluid findings Brain MRI normal in 70% of patients Angiographic findings similar to PACNS but usually reversible within 6–12 weeks No inflammatory changes on the brain biopsy
Multiple sclerosis [10].	Less contrast-enhancing lesions (compared to PACNS in which 90% of lesions enhance contrast) No vessel wall changes
Anti-MOG encephalitis [23].	Positive MOG antibodies in the CSF No vessel wall changes Absence of fibrinoid necrosis of the vessel wall on the brain biopsy
Sneddon’s syndrome [16].	Younger women are affected (20–40 y.o.) Livedo rash Antiphospholipid antibodies positive in 50–80% Headache is not so common Psychiatric complications occur later in the course of the disease

PACNS—primary angiitis of the central nervous system; ESR—erythrocyte sedimentation rate; CRP—C-reactive protein; MRI—magnetic resonance imaging; MOG—myelin oligodendrocyte glycoprotein; CSF—cerebrospinal fluid.

**Table 3 medicina-58-01446-t003:** The most common features of primary CNS vasculitides and their differences.

	PACNS	CAARI	ABRA
Cognitive decline	50–70% [17]	48–76% [25,28]	71% [29]
Focal neurological symptoms	25–30% [17]	46% [25,28]	51% [29]
Severe headache	50–60% [17]	32–41% [25,28]	35% [29]
Seizures	25–30% [17]	31% [25,28]	30% [29]
Elevated ESR or CRP [31]	Uncommon	Up to 30%	
Elevated CSF protein [31]	80–90%	71%	
CSF pleocytosis [31]	80–90%	44%	
Intracranial hemorrhage [31]	8%	20%	14%
Meningeal gadolinium enhancement [31]	12%	56%	57%

PACNS—primary angiitis of the central nervous system, CAARI—cerebral amyloid angiopathy related inflammation, ABRA—amyloid β-related angiitis, ESR—erythrocyte sedimentation rate, CRP– C-reactive protein, CSF—cerebrospinal fluid.

**Table 4 medicina-58-01446-t004:** CNS involvement in systemic vasculitides.

Disorder	Frequency of CNS Involvement (%)	Disorder	Frequency of CNS Involvement (%)
Systemic lupus erythematosus [36,37]	30–50%	Granulomatosis with polyangiitis [38]	7–11%
Giant cell arteritis [33]	30%	Eosinophilic granulomatosis with polyangiitis [38]	6–10%
Takayasu arteritis [38]	10–20%	Henoch-Schoenlein purpura [38]	0.65–8%
Polyarteritis nodosa [38]	5–25%	Kawasaki disease [38]	5.1%
Microscopic polyangiitis [36]	11%	Behçet’s disease [39]	3–30%

CNS—central nervous system.

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
