# Peer review of "Inflammatory Disorders of the Central Nervous System Vessels: Narrative Review"

_medicina, 2022, doi:10.3390/medicina58101446_

Round 1

Reviewer 1 Report (Previous Reviewer 1)

I want to congratulate the authors for their significant improvement in the quality of the manuscript.

1. The title should describe the type of review. E.g., Narrative.

2. Revise authors' filiation. There are three numbers, but only two filiations.

3. There are some typos and misspellings throughout the manuscript.

Author Response

  1. Corrected
  2. Revised
  3. Revised and corrected.

Reviewer 2 Report (Previous Reviewer 3)

- The heading "Discussion" on page 23 should be replaced with "Diagnosis of the Inflammatory Disorders of the Central Nervous System Vessels".

- "Materials and methods" mentioned at the end of the manuscript on page 24 should be transferred to the end of the introduction after the paper's aim.

Author Response

  1. This part was called "Future directions in the workup of inflammatory disorders of the central nervous system vessels", as there was earlier request from the reviewers to put a paragraph about future directions.
  2. Corrected.

This manuscript is a resubmission of an earlier submission. The following is a list of the peer review reports and author responses from that submission.

Round 1

Reviewer 1 Report

1. The title should describe the type of review. E.g., Narrative.

2. The authors should provide more specific data in the abstract. Remember that in most indexations, only the abstract is provided.

3. The methodology should be uploaded as supplementary material.

4. References 16, 17, 29, and 30 are incomplete. It is advised to make necessary adjustments.

5. A figure about the mechanism of vasculitides would significantly impact the quality of the manuscript.

6. The first appearance of every term should have its full description. E.g., GRE, SWI.

7. Table 1 is advised to increase the number of variables to lead to a better distinction among the primary vasculitides.

8. Could the authors provide a figure- or video abstract?

9. A chapter about future directions should be provided.

10. A specific chapter with a figure or table with hints for diagnosing the vasculitides is essential.

Reviewer 2 Report

Title Inflammatory Disorders of the Central Nervous System Vessels

The authors carried out a review article on inflammatory disorders of the central nervous system vessels. this manuscript adds, but not so substantially to the existing literature. i find it to be acceptable although there are a few comments that must be addressed.

·       English editing is mandatory

·       Abstract

·       The writing language is in need to be more smooth

·       Line 26 ...that …this sentence is not appropriate

·       Line 31 ... Brain biopsy and precise diagnostics ...what is meant by diagnostics

·       Introduction ...there are no references at the introduction ...which is not appropriate

·       Line 37 the authors starts with enumerating the types ..however no tiltle to indicate that the coming subtitles is for the types or causes

·       Line 42 ..while other ..again English editing is needed

·       Line 55..the most common symptoms …these are not symptoms ..The authors means manifestations or presentation

·       Line 60.. other 40% visit the doctor  within 4 weeks …not appropriate sentence

·       Paragraph 54 ..paragraph 60 …There are repetition of the same ideas

·       Paragraph 74 ..the same idea of the previous 2 paragraphs

·       Paragraph 84 ..the starting and closing sentences are to some extent the same

·            Line 149.. (starting at 750 mg/m2 )…m2

·       line 213..the manuscript is deficient in any images that illustrates the idea of the authors especially for the MRI CNS pattern

·       line 238 .. prednisolone 1mg/kg/p, what is meant by P

·            Line 240 .. 750 mg/m2..m2

·       Line 243 .. relapsean..what is meant by this word 

Reviewer 3 Report

This review article described clinical features and course, current diagnostic workup and treatment guidelines of the non-infectious CNS vasculitides. There are some concerns in this manuscript as follows:

- Affiliations: The country in which Vilnius University is present should be mentioned.

- Introduction: References should be added to the introduction. It is written without mentioning any references.

- The classification and the predisposing factors of the inflammatory disorders of CNS vessels should be mentioned.

- A guide to the differential diagnosis of the inflammatory disorders of CNS vessels should be mentioned.

- A figure summarizing the pathogenic mechanisms of the inflammatory disorders of CNS vessels is recommended.

- The manuscript should be revised regarding the spelling and the grammatical errors.